# STORM: Sketch Toward Online Risk Minimization

## Abstract

Empirical risk minimization is perhaps the most influential idea in statistical learning, with applications to nearly all scientific and technical domains in the form of regression and classification models. The growing concerns about the high energy cost of training and the increased prevalence of massive streaming datasets have led many ML practitioners to look for approximate ERM models that can achieve low cost on memory and latency for training. To this end, we propose STORM, an online sketching-based method for empirical risk minimization. STORM compresses a data stream into a tiny array of integer counters. This sketch is sufficient to estimate a variety of surrogate losses over the original dataset. We provide rigorous theoretical analysis and show that STORM can estimate a carefully chosen surrogate loss for regularized least-squares regression and a margin loss for classification. We perform an exhaustive experimental comparison for regression and classification training on real-world datasets, achieving an approximate solution with size even less than a data sample.

## 1 Introduction

Empirical risk minimization (ERM) has been independently studied by the machine learning, statistics, and computer science communities. However, conventional ERM algorithms are becoming more expensive in terms of latency, memory, and energy due to the need for accurate inference over rapidly increasing data volumes (He et al., 2016; Vaswani et al., 2017). The net effect is an increase in storage costs of large data matrices (Guhaniyogi & Dunson, 2015), network latency for personalized inference/recommendations, and carbon emissions. There is a growing need for approximation algorithms that can minimize the cost, memory, and energy associated with training and inference (Lin et al., 2020; Motamedi et al., 2016). Additionally, there are distinct advantages of training on a small memory local/personalized device, as data is generally collected on these devices. This limits the data transfer to big servers that consumes substantial resources (Bill Allcock, 2001; Xu, 2013) and minimizes the privacy vulnerabilities (Mendez Mena et al., 2018). Existing compression methods generally fall into two categories: sketching and sampling.

**Linear Sketches:** A large family of projection-based sketches uses ideas from compressed sensing, matrix factorization, and dimensionality reduction. A common theme is to approximate the large $N \times d$ data matrix with a smaller representation. For example, multiplication with a random $\pm 1$ matrix yields a sketch that attains the information-theoretic lower bound for compressed linear regression (within a constant factor) (Clarkson & Woodruff, 2009). Random Gaussian, Hadamard, Haar, and Bernoulli sign matrices have also been used in this framework with varying degrees of theoretical and practical success (Dobriban & Liu, 2019). The "sketch-and-solve" technique has also been extended to regression with the $\ell^1$ objective and several other linear problems (Sohler & Woodruff, 2011). This framework cannot accommodate regularized objectives because the model is computed directly from a set of precomputed sufficient statistics.

Recent work by Shi & Phillips (2021) has shown that a matrix sketch based on the frequent directions method (Liberty, 2013; Ghashami et al., 2016) can be used for ridge regression. This algorithm uses a sketch to approximate the most costly part of the ridge regression objective, $\mathbf{X}^T\mathbf{X} + \lambda\mathbf{I}$, in a streaming setting. The memory is $O(ld)$, where $\mathbf{X} \in \mathcal{R}^{n \times d}$ and $l > 2$. That is, the memory is linear in the dimensions, with a minimum size of $(2d + 1) \times$ floating precision that can be very large for high-dimensional data.

**Coresets and Sampling:** A reasonable strategy is to extract a core subset (or *coreset*) of points that efficiently represent the complete dataset. The coreset framework supports the approximation of loss function sums within a $(1 \pm \epsilon)$ error margin. There are coresets for important problems such as $\ell^p$-regression (Dasgupta et al., 2009) and logistic regression (Munteanu et al., 2018). Classification objectives also admit small coresets (Har-Peled et al., 2007; Tukan et al., 2020), as do many other regularized ERM problems (Reddi et al., 2015). The problem is that many coresets require super-linear construction time. For example, many of the algorithms in the previous paragraph require k-means clustering, leverage scores, or multiple cycles through the dataset (Tukan et al., 2020; Munteanu et al., 2018; Har-Peled et al., 2007). Coresets are also tricky to merge since it is not straightforward to decide which points to discard. As a result, *composable* and *streaming* coresets are an active area of research (Indyk et al., 2014; Karnin & Liberty, 2019). Although leverage scores can be approximated online (Cohen et al., 2016), the process is computationally expensive and not well suited to edge devices. In many practical problems, random sampling is the only feasible sampling baseline that supports regularized ERM.

In this work, we propose a new sketching framework (STORM) that supports both regularization and streaming algorithms. Given an $N$-point dataset $\mathcal{D} = \{(x_1, y_1)...(x_N, y_N)\}$ of $d$-dimensional examples observed in the streaming setting (Fiat, 1998), the goal is to construct a small structure $\mathcal{S}$ that can estimate a regression and classification loss over $\mathcal{D}$. STORM consists of two main components, 1) STORM loss for regression and classification, 2) A tiny array of integer count values called STORM sketch. This array is sufficient to perform empirical risk minimization on STORM loss. In practice, our data structure allows for optimizations that are not possible with other methods. Specifically, our contributions are as follows:

1. We propose an online sketching algorithm (STORM) that can estimate a special class of loss functions using only integer count values. Our sketch is appropriate for distributed data settings because it is small, embarrassingly parallel, and mergeable via addition.

2. We characterize the set of STORM-approximable loss functions. The price of efficient loss estimation is a restriction on the kinds of losses we can use. While we cannot directly estimate popular regression and classification losses, we construct surrogate losses for $\ell^2$ regression and large margin classification.

3. We show how to optimize models over STORM sketches. The count-based nature of our sketch makes gradient descent difficult, and black-box optimization techniques are of limited use in high-dimensional spaces. We propose an optimization algorithm that is custom-tailored to our sketch.

4. We conduct a rigorous empirical evaluation with ridge regression and classification. We show that our approach has better train/test loss performance and lower variance than sampling baselines in the low-memory, resource-constrained computational regime. For some datasets, we can achieve strong performance with memory smaller than even one sample.

## 2 BACKGROUND

### 2.1 EMPIRICAL RISK MINIMIZATION

In the standard statistical learning framework, we are given a training dataset $\mathcal{D} = \{(x_i, y_i) \in \mathcal{D} \times \mathcal{Y}\}_{i=1}^{n}$ of examples and asked to select a function $h : \mathcal{X} \to \mathcal{Y}$ that can predict $y$ given $x$. In this paper, we will present our algorithms for the data space $\mathcal{X} = \mathbb{R}^d$ and output space $\mathcal{Y} = \mathbb{R}$, but most of our results hold generally for other metric spaces too. The learning problem is to select a hypothesis $h_\theta$ (a function parameterized by $\theta$) that yields good predictions, as measured by a loss function $\text{loss}(h_\theta(x), y)$. In empirical risk minimization (ERM), we select $\theta$ to minimize the average loss on the training set.

$$\hat{\theta} = \arg\min_{\theta} \sum_{i=1}^{n} \text{loss}(h_\theta(x_i), y_i)$$

In the case of **Linear regression**, $h_\theta(\mathbf{x}) = \langle \theta, \mathbf{x} \rangle$. For most applications, $\theta$ is found using the least-squares or $\ell^2$ loss: $\text{loss}(h_\theta(\mathbf{x})) = ||h_\theta(\mathbf{x}) - y||_2^2$. The unconstrained $\ell^2$ loss is smooth and

strongly convex, with favorable convergence criteria. The parameter $\theta$ is typically found using gradient descent or via a closed-form solution from the matrix formulation of the model $\mathbf{y} = \mathbf{X}\theta$. For our discussion, it will be important to express the loss in terms of the concatenated vector $[\mathbf{x}_i, y_i]$:

$$\hat{\theta} = \arg\min_{\theta} \sum_{i=1}^{n} (\langle [\mathbf{x_i}, y_i], [\theta, -1] \rangle)^2 = \arg\min_{\theta} \|\mathbf{y} - \mathbf{X}\theta\|_2^2$$

## 2.2 LOCALITY SENSITIVE HASHING

A Locality sensitive hash (LSH) function ($l(\mathbf{x})$) provides the same hash value for similar points with high probability. The notion of similarity is usually based on the distance measure of the metric space $\mathcal{X}$. For instance, there are LSH families for the Jaccard (Broder, 1997), Euclidean (Datar et al., 2004; Dasgupta et al., 2011) and angular distances (Charikar, 2002). If we allow asymmetric hash constructions (i.e. $l_1(\mathbf{x_1}) = l_2(\mathbf{x_2})$, $l_1 \neq l_2$), then $\mathbf{x}_1$ and $\mathbf{x}_2$ may collide based on other properties such as their inner product (Shrivastava & Li, 2014). Asymmetric LSH functions use one hash function ($l_1$) for data and a different function ($l_2$) for queries. To accommodate asymmetric functions, we use the following definition of LSH, Our definition is a strict generalization of the original definition from Indyk (1998) Indyk & Motwani (1998), which can be recovered by setting probability thresholds.

**Definition 1.** *We say that a hash family $\mathcal{F}$ is locality-sensitive with collision probability $k(\cdot, \cdot)$ if for any two points $\mathbf{x}_1$ and $\mathbf{x}_2$ in $\mathbb{R}^d$, $l(\mathbf{x}_1) = l(\mathbf{x}_2)$ with probability $k(\mathbf{x}_1, \mathbf{x}_2)$ under a uniform random selection of $l(\cdot) \in \mathcal{F}$.*

The **signed random projection (SRP)** family for the angular distance (Goemans & Williamson, 1995; Charikar, 2002) is an example of a symmetric LSH family. The SRP family is the set of functions $l(\mathbf{x}) = \text{sign}(\mathbf{w}^\top \mathbf{x})$, where $\mathbf{w} \sim \mathcal{N}(0, I_d)$. The SRP collision probability is

$$k(\mathbf{x}_1, \mathbf{x}_2) = 1 - \frac{1}{\pi} \cos^{-1} \left( \frac{\langle \mathbf{x}_1, \mathbf{x}_2 \rangle}{\|\mathbf{x}_1\|_2 \|\mathbf{x}_2\|_2} \right)$$

The inner product hash Shrivastava & Li (2014) is an example of an asymmetric LSH. To get this hash function, replace $\mathbf{x}_1$ with $\left[\mathbf{x}_1, 0, \sqrt{1 - \|\mathbf{x}_1\|_2^2}\right]$ and $\mathbf{x}_2$ with $\left[\mathbf{x}_2, \sqrt{1 - \|\mathbf{x_2}\|_2^2}, 0\right]$ in the SRP function. This procedure essentially uses different hash functions for $\mathbf{x}_1$ and $\mathbf{x}_2$, since we augment $\mathbf{x}_1$ (the data) in one way and $\mathbf{x}_2$ (the query) in a different way. The collision probability $k(\mathbf{x}_1, \mathbf{x}_2)$ is now a monotone function of the inner product $\langle \mathbf{x}_1, \mathbf{x}_2 \rangle$. This construction implicitly assumes that $\mathbf{x}_1$ and $\mathbf{x}_2$ are inside the unit sphere, but other hash functions do not require this assumption (Shrivastava & Li, 2015; Neyshabur & Srebro, 2015; Huang et al., 2018).[1]

**RACE Sketch:** The RACE sketch is a 2D integer array $\mathcal{S} \in \mathbb{Z}^{R \times B}$ that we index using LSH functions (Coleman & Shrivastava, 2020; Luo & Shrivastava, 2018). Each row of $\mathcal{S}$ is called an Array-of-Counts (ACE) data structure (Luo & Shrivastava, 2018), and the entire sketch is a set of $R$ independent ACE repetitions (RACE). To construct a RACE sketch from a dataset $D$, we begin with an empty $R \times B$ integer array and a set of $R$ independent LSH functions $\{l_1(\mathbf{x})...l_R(\mathbf{x})\}$. Each LSH function has the collision probability $k(\theta, \mathbf{x})$. When an element $\mathbf{x}$ arrives from the data stream, we hash $\mathbf{x}$ to get $R$ different hash values. We increment row $i$ at location $l_i(\mathbf{x})$ and repeat this process for all elements in the dataset.

To *query* the sketch with a query $\theta$, we return the average value of $\mathbf{S}[i, l_i(\theta)]$ over the $R$ rows. RACE sketches are interesting because they estimate the sum of LSH collision probabilities over the dataset with low variance for any query $\theta$.

**Theorem 1.** *Unbiased RACE Estimator Coleman & Shrivastava (2020) Let $X = \mathcal{S}[i, l_i(\theta)]$. Then*

$$\mathbb{E}[X] = \sum_{\mathbf{x} \in \mathbf{D}} k(\mathbf{x}, \theta) \qquad \text{var}(X) \leq \left( \sum_{\mathbf{x} \in \mathcal{D}} \sqrt{k(\mathbf{x}, \theta)} \right)^2$$

---

[1] Regardless, we often scale and center the dataset when using inner product hash functions in practice.

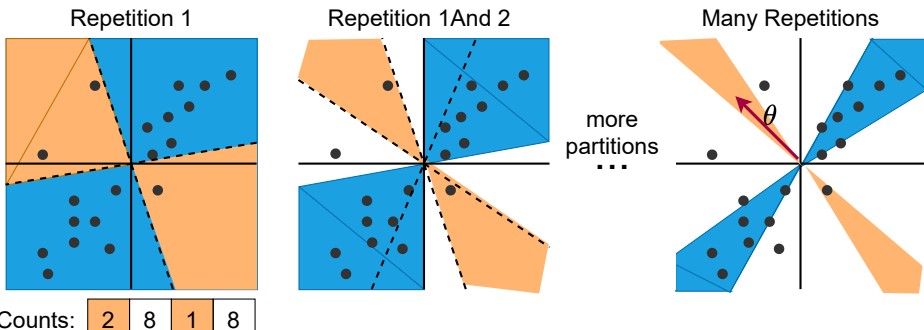

Figure 1: LSH counts are sufficient for learning regression lines and classification hyperplanes. Here, we show the partitions that correspond to several STORM repetitions (or rows). Since the regression line should lie inside a densely-populated region, we use sparsely-populated partitions (shown in brown) to identify regions that contain the normal to the regression line (regression hyper-plane in higher dimensions). Each STORM repetition prunes the space of eligible models, until we are left with the small feasible sparse region (shown in orange). We gain the most information when the partition boundaries are orthogonal, which is nearly always true in high dimensions.

## 3 STORM

While collision probabilities have been well-studied in the context of improving near-neighbor search (Gionis et al., 1999), we study them from a new perspective. Our goal is to compose useful loss functions from LSH collision probabilities and perform ERM on sketches. In this context, we query the sketch with the parameter $\theta$ to estimate the empirical risk. This requires some extensions to the RACE sketch.

**Intuition** To understand our approach, consider the task presented in Figure 1. We partition the data space into $B$ random regions using Gaussian random hyperplanes. The count of points in each region gives an estimate of its density. We repeat this partitioning many times so that we have a good estimate of densely-populated and sparsely-populated regions. In the regression settings, the optimal model hyperplane passes through the dense regions. Our objective is to find a $\theta$ vector normal to the optimal model hyperplane. This motivates to look for the sparsest region. STORM sketch (inspired from RACE) provides counts of data points in each partition. We will see that we can optimize over these counts to find a $\theta$ that resides in the sparsest region. The main challenge is to design an LSH function $l(\theta)$ that has the minimal count when $\theta$ is a good model. Next, we will talk about an appropriate hash function, STORM loss for regression and classification and, the STORM sketch for approximating the loss.

### 3.1 STORM SURROGATE LOSS FOR LINEAR REGRESSION

Since we query the STORM sketch with $\theta$, we must be able to express $\text{loss}(h_\theta(x), y)$ as $\text{loss}(\theta, [x, y])$. This condition is not limiting, but it does restrict our attention to hypothesis classes where $\theta$ directly interacts with $x$ and $y$ in a way that can be captured by an LSH function. Linear regression is the simplest example of such a model. Here the $\theta$ interacts with the data via the inner product $\langle [\theta, -1], [\mathbf{x}, y] \rangle$. The asymmetric hash function from Section 2.2 seems like a natural choice. However, the linear regression loss function is a monotone function of the *absolute value* of the inner product.

To obtain a surrogate loss with the correct dependence on $\langle [\theta, -1], [\mathbf{x}, y] \rangle$, observe that the collision probability of SRP (Section 2.2) is monotone decreasing if we hash $[-\mathbf{x}, -y]$ instead of $[\mathbf{x}, y]$. Thus, we can obtain a function that is monotone in $|\langle [\theta, -1], [\mathbf{x}, y] \rangle|$ by adding together the collision probability for $[\mathbf{x}, y]$ (which is monotone increasing) and for $[-\mathbf{x}, -y]$ (which is monotone decreasing). We call this LSH family "**paired random projections**" (PRP) because the method can be implemented by hashing both $[\mathbf{x}, y]$ and $[-\mathbf{x}, -y]$ with a given SRP function and updating both locations in the sketch. When we query the vector $\tilde{\theta} = [\theta, -1]$ with the same SRP function, STORM estimates the following collision probability/ **surrogate loss for linear regression**

$$g(\tilde{\theta}, [\mathbf{x}, y]) = \frac{1}{2}\left(1 - \frac{1}{\pi}\cos^{-1}(\langle \tilde{\theta}, [\mathbf{x}, y] \rangle)\right)^p + \frac{1}{2}\left(1 - \frac{1}{\pi}\cos^{-1}(-\langle \tilde{\theta}, [\mathbf{x}, y] \rangle)\right)^p,$$

where we are free to choose any number of random hyperplanes - $p \geq 2$. The proposed loss is convex, unbiased and consistent. More info in section 4.

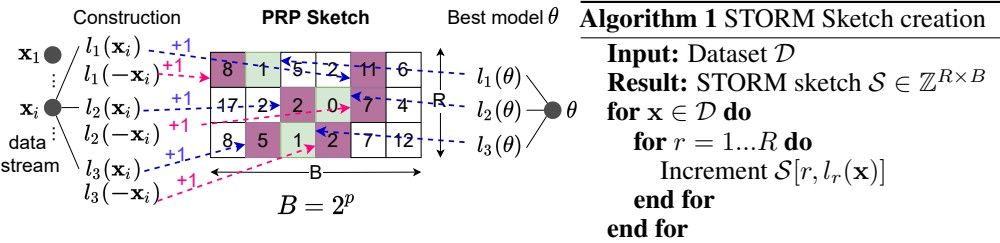

Figure 2: To construct a STORM sketch, we initialize a 2D array with $R$ rows and $B$ columns (buckets) in each row. The asymmetric LSH function $l_i$ indexes row $i$ of the sketch. We want best model to hash into least count buckets. See Sections 3.3 for details.

## 3.2 STORM SURROGATE LOSS FOR CLASSIFICATION

Consider linear hyperplane classifiers of the form $h_\theta(x) = \text{sign}(\langle \theta, [\mathbf{x}, -1] \rangle)$. The most popular losses used to find $\theta$ are classification-calibrated margin losses. A loss function is classification-calibrated, or Bayes consistent, if the optimal hypothesis under the loss is the same as the Bayes optimal hypothesis. With these factors in mind, we propose **Classification Random Projections** (CRP) to estimate the following loss function:

**Theorem 2.** *Consider labels $y \in \{-1, 1\}$. The loss function $g(\theta, [\mathbf{x}, y])$ for the linear hyperplane classifier is a classification-calibrated margin loss and is STORM-approximable.*

$$g(\theta, [\mathbf{x}, y]) = 2^p \left( 1 - \frac{1}{\pi} \cos^{-1}(-y\langle \theta, \mathbf{x} \rangle) \right)^p$$

By optimizing $\theta$ in a similar fashion to as with regression, we can train linear classifiers with STORM sketches. The LSH function that implements $g(\tilde{\theta}, [\mathbf{x}, y])$ is the asymmetric inner product (SRP) hash, but with the argument to the hash function multiplied by $y \in \{-1, +1\}$.
See the appendix for details and proof of theorems.

## 3.3 STORM SKETCH

We use asymmetric LSH functions to hash data points $\mathbf{x}$ and $\theta$ with different functions. Most LSH functions have positive definite collision probabilities, but asymmetric LSH functions allow much more flexibility. Our Paired Random Projections (PRP) hashing scheme premultiply the data vector by a $\{\pm 1\}$ to estimate a surrogate regression loss. Classification Random Projections (CRP) LSH premultiplies $\mathbf{x}$ with $\mathbf{y}$. Figure 2 shows our sketch process for PRP. Here we insert elements into the sketch with two SRP hashes, but only use one SRP for the query. In Section 3.1 and Section 3.2, we show that this sketch estimates a surrogate loss for linear regression and classification.

Once we have a sketch and an appropriate hash function, we want to optimize the model parameter $\theta$ to minimize the STORM estimate of the empirical risk. Proposed optimization method is discussed in details in the next section.

## 3.4 OPTIMIZATION

Gradient descent is traditionally used to find the best $\theta$ for ERM problems. Unfortunately, our sketch consists of discrete integer counts that have neither an analytic gradient nor a subgradient. However, following the intuition from Figure 1, we know that our surrogate loss tries to find the hyperparameter $\tilde{\theta}$ in the sparsest region. This allows us to develop a specialized optimization heuristic that exploits the hyperplanes of the proposed hash to locate the minimum-risk $\theta$ more effectively. Our surrogate objective is: $\hat{\theta} = \arg\min_\theta g([\theta, -1], [\mathbf{x}, y]) = \arg\min_\theta \mathbb{E}(\mathcal{S}[L([\theta, -1])])$.

**Algorithm 2** Hyperplane-based stochastic gradient descent

**Input:** STORM sketch $\mathcal{S} \in \mathbb{Z}^{R \times B}$ **Result:** Model $\theta$
Initialise $\tilde{\theta}_0 \leftarrow \mathbf{0}^{d+1}$
**for** $i = 1..R$ **do**
  $Z_i = 2 * \text{binary}(\arg\min \mathcal{S}[i]) - 1$
**end for**
Aggregate all labels $Z_i's$ to $Z$ and store it.
**for** iterations $t = 1...T$ **do**
  random sample $i$
  Get random hyper-planes $W_i$ from the given seed.
  Update $\tilde{\theta}_t^j = \tilde{\theta}_{t-1}^j - \eta \nabla \|\tanh(W_i^\top \tilde{\theta}) - Z_i\|_2^2 - 2\lambda \tilde{\theta}$
  Project last dimension of $\tilde{\theta}$ to $-1$
**end for**
Select best $\tilde{\theta}^j$ using a validation set.

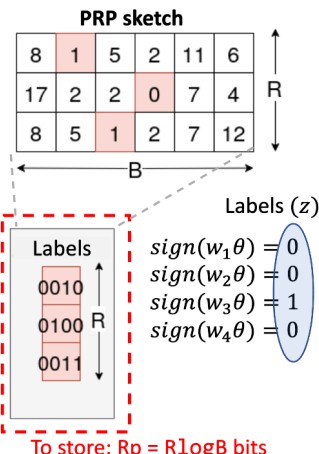

Figure 3: Left: Shows the algorithm for optimization on the extracted labels of least count bins. Right: Figure that shows the further compression on the PRP sketch (works the same for CRP) by storing just the labels ($Z$) of least count buckets.

where $\mathcal{S}$ is the sketch and $L = \{l_1..l_p\}$ is the hash function. This implies that $\theta$ lies in a region where most of the STORM partitions have lower count values (small values of $\mathcal{S}[l(\tilde{\theta})]$). One could do a brute-force minimization over all possible partitions and select a model $\tilde{\theta}$ in the winning partition, but this is computationally expensive. Instead, we use the stochastic gradient descent approach and transform our objective as follows: $\hat{g}(\theta) = \sum_{i=1}^R \mathcal{S}[i, L_i([\theta, -1])]$

The gradient is $\nabla \hat{g}(\theta) = \sum_{i=1}^R \nabla \mathcal{S}[i, L_i([\theta, -1])]$.

It turns out that the individual minimization problem (for each $i^{th}$ sample), $\arg\min_\theta \mathcal{S}[i, L_i([\theta, -1])]$ can also be written as $\arg\min_\theta \|L_i([\theta, -1]) - Z_i\|_2^2$. Here $Z_i \in \{0, 1\}^p$ represents the binary label vector of the minimum count bucket in row $i$.

Because we use SRP as the hash function, $L_i(\tilde{\theta}) = \text{sign}(W_i^\top \tilde{\theta})$. We now have the objective function in terms of $\frac{1}{2}|\text{sign}(W_i^\top \tilde{\theta}) - Z_i\|_2^2$. This can be made diffrentiable by using $\tanh$ instead of $\text{sign}$ function. Hence, the gradient update becomes

$$\tilde{\theta}^{t+1} = \tilde{\theta}^t - \eta \nabla \|\tanh(W_i^\top \tilde{\theta}) - Z_i\|_2^2$$

This becomes the SGD over a standard single layer neural net with $\tanh$ activation. (Refer to appendix for more details). Also, we can simply add any regularization term with this. In case of Ridge regression we have the following gradient update

$$\tilde{\theta}^{t+1} = \tilde{\theta}^t - \eta \nabla \|\tanh(W_i^\top \tilde{\theta}) - Z_i\|_2^2 - 2\lambda \tilde{\theta}$$

It is interesting to note that, we are learning over randomly generated hyperplanes ($W_i$) (these only need a single seed value for generation) and very cheap binary vectors $Z \in \{0, 1\}^{R \times p}$. The sketch size required is $Rp$ or $R \log B$. Although this method (Algorithm 2) finds approximate gradients, we find that the procedure works well in practice. Since we can regenerate the hyperplanes from random seeds, we only need to store the hash codes ($Z$) of the least populated buckets with this approach.

## 4 THEORY: SKETCHABLE SURROGATE LOSS

We begin with a formal description of the families of losses that STORM can approximate.

Formally, a loss $g(\theta, [x, y])$ is **STORM-approximable** if there is a hash function and a STORM sketch such that querying the STORM sketch with $\theta$ yields an unbiased estimator of $g(\theta, [x, y])$.

Using the compositional properties of LSH, we can approximate any sum and/or product of LSH collision probabilities using one or more STORM sketches. For example, we estimate sums by

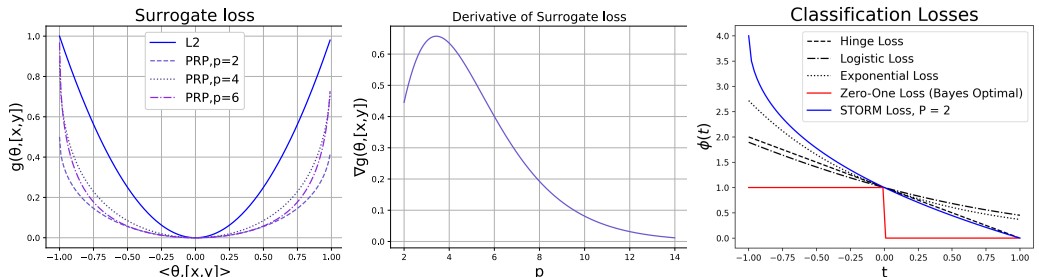

Figure 4: (a) PRP Surrogate loss for different values of $p$. It is convex, but less sharp than the L2 loss. A key observation is that $p = 4$ produces the highest convexity among the family of PRP surrogate losses. (b)Plot of slope surrogate loss for different values of $p$ at $\langle [\theta, -1], [\mathbf{x}, y] \rangle = 0.1$. The surrogate loss has the steepest slope (and is thus easiest to optimize) near $p = 4$. (c) Comparison of various classification-calibrated loss functions, including the STORM's CRP loss.

inserting elements with more than one LSH function (or using a second sketch). We can estimate a product of collision probabilities by combining multiple LSH outputs into a single hash code. Theorem 3 formalizes this argument (we defer all proofs to the appendix).

**Theorem 3.** *The set of STORM-approximable functions $S_L$ contains all LSH collision probabilities and is closed under addition, subtraction, and multiplication.*

Given the number of known LSH families, the flexibility of asymmetric LSH and the closure of $S_L$ under addition and multiplication, Theorem 3 suggests that $S_L$ is an expressive space of functions. It remains for us to show that $S_L$ contains useful losses for machine learning problems.

**Theorem 4.** *When $p \geq 2$, the PRP collision probability $g([\theta, -1], [\mathbf{x}, y])$ is a convex surrogate loss for the linear regression objective such that*

$$\arg\min_\theta g([\theta, -1], [\mathbf{x}, y]) = \arg\min_\theta \|y - \mathbf{x}\theta\|$$

Check Figure 4 (a). A detail proof is given in the Appendix.
We analyse, how the sketch size affects the PRP/CRP loss estimation from the STORM sketch. Consider the following **Axiom**
*Let $Z_1, ... Z_R$ be $R$ i.i.d. random variables with mean $\mathbb{E}[Z] = \mu$ and variance $\leq \sigma^2$. Divide the $R$ variables into $g$ groups so that each group contains $m = R/g$ elements, and take the empirical average within each group. The median-of-means estimate $\hat{\mu}$ is the median of the $g$ group means. If $g = 8\log(1/\delta)$ and $m = R/g$, then the following statement holds with probability $1 - \delta$.*

$$|\hat{\mu} - \mu| \leq 6\frac{\sigma}{\sqrt{R}}\sqrt{\log 1/\delta}$$

This proof is given in Alon et al. (1999) as the proof of Theorem 2.1 (which is a slightly more general version of the statement above). From the variance result from Theorem 1, and a median of means estimate of the PRP/CRP loss from STORM sketch, we state the following-

**Theorem 5.** *Let $\hat{g}(\theta)$ is the median of means estimate on the STORM sketch, with $R$ rows, then with probability more than $1 - \delta$*

$$|\hat{g}(\theta) - g(\theta)| \leq 6\frac{\sum_{\mathbf{x} \in \mathcal{D}} \sqrt{k(\mathbf{x}, \theta)}}{\sqrt{R}}\sqrt{\log(1/\delta)}$$

Where $\sum_{\mathbf{x} \in \mathcal{D}} \sqrt{k(\mathbf{x}, \theta)}$ is the collision probability of the hash function in use.

## 4.1 PARAMETERS

**Number of hash functions (p):** The parameter $p$ is the number of PRP/CRP hash hyperplanes that creates $2^p$ partitions. Larger $p$ is good for more resolution of the space, but it also leads to many buckets with few/zero counts. (Figure 4(a) shows that higher $p$ makes the loss surface flatter near the optimal region). Figure 4 (b) shows that setting $p \approx 4$ maximizes the gradient of the surrogate loss near the minima, and we find that in practice too (Refer Appendix). It is also efficient because for $p = 4$, $B$ is only 16. Also, we need fewer rows in the sketch to optimize $g(\tilde{\theta}, [\mathbf{x}, y])$ when it is strongly convex.

Table 1: Each dataset has $N$ entries and $d$ features. The datasets are taken with Open Commons share-alike license and Creative Commons Attribution License. The dimension in bracket represent the effective dataset dimension after random Fourier feature mapping.

| Dataset | $N$ | $d$ | Size | Type |
|---|---|---|---|---|
| Diabetes (Efron et al., 2004) | 442 | 10 | 17KB | Patient records |
| Boston (Harrison & Rubinfeld, 1978) | 500 | 13 | 25KB | Housing price |
| Meat (Jowder et al., 1997) | 239 | 123 (615) | 114KB | Mid-IR spectra |
| Soil (Corwin, 2019) | 679 | 127 (635) | 336KB | Sensor data |
| Gas (Vergara et al., 2012) | 13910 | 129 | 6.8MB | Sensor data |
| Fruit (Holland et al., 1998) | 983 | 235 (1175) | 902KB | Mid-IR spectra |
| UJIndoorLoc (Torres-Sospedra et al., 2014) | 21048 | 529 | 42MB | Localization |
| Australian (Dua & Graff, 2017) | 690 | 14 | 37KB | Financial data |
| Mnist (Lecun et al., 1998) | 60000 | 784 | 179MB | Image data |

**Number of repetitions (R):** The square root of collision probability $\sqrt{k(\mathbf{x}, \theta)} \leq 1$ always. Hence, in the worst case the value of $\sum_{\mathbf{x} \in \mathcal{D}} \sqrt{k(\mathbf{x}, \theta)}$ is $N$. For $|\hat{g}(\theta) - g(\theta)| \leq C\sqrt{\log(1/\delta)}$, we need number of sketch rows $R < O(N^2)$ (Using the Theorem 5). The worst-case $O(N^2)$ situation happens when every point in the dataset has maximum loss. Practical modeling situations typically have a small fraction of points for which the training loss is small. In experiments, we vary the number of rows and see the change in Mean Square Error (MSE).

## 5 EXPERIMENTS

**Experiment setup:** We use PRP and CRP with $p = 4$ (most optimal, check Figure 4(b)) to create the sketch and we vary the number of repetitions to show the memory tradeoff. We have used a variety of datasets. Refer table 1

All models are trained and evaluated with a fixed 80/20 train/test split. We average over 100 runs for our baselines and sketches, where each run has an independently constructed sketch or random sample. Thus, our average is over the random LSH functions used to construct the sketch and the stochastic gradient descent instances. We use the hyperplane optimization heuristic for regression and derivative-free (Conn et al., 2009) gradient descent for classification. To run the code we used our personal laptop with i7-8750H processor, 2.2 GHz base frequency and 9MB cache size.

**Ridge Regression:** We evaluate our method on ridge regression, which penalizes $\theta$ to avoid the double descent and over-fitting problem (Nakkiran, 2019). To implement this model, we simply add $\|\theta\|_2^2$ to the STORM objective function. The gradient updates take place as described in Section 3.3, but with an added regularization term. For regularized ERM in our computational setting, random sampling and Frequent direction sketch is the only reasonable baseline. Coresets for ridge regression exist, but cannot be constructed in the streaming setting. Also, Frequent direction based sketch takes space much more than STORM and random samples for higher-dimensional data (near $d > 100$). Hence a direct comparison is made for the low dimensional- Diabetes and Boston housing dataset. Figure 5 shows the results of our comparison focused mostly on sub-kB region. We find that STORM achieves a lower MSE in the low-memory regime. In particular, for the gas, soil, fruit, and indoor datasets, STORM produces a reasonable model using less memory than even a single sample. Finally, the STORM results have a substantially smaller variance (typically 2x), which is a critical factor in practice. Additional plots and sanity check on a 2D synthetic data is available in the appendix.

**STORM for Non-linear Objectives:** STORM supports regularized linear regression and classification. With this formulation, a non-linear model can be trained with feature mapping such as random Fourier features (Rahimi & Recht, 2008). To achieve a non-linear boundary the mapping is done to a higher dimension space. Practically this is $3 - 6$ times the original dimensions. Since STORM does well with high-dimensional datasets, often requiring less memory than a single sample, we expect good performance using high–dimensional feature transformations. Figure 5 last row demonstrate the mean square error (MSE) with the size of sketch variables required for linear regression.

**Linear classification:** We run the linear classification on datasets from the libsvm repository. We compare against a linear SVM trained on a random sample of the dataset. Figure 5 shows this comparison for the "Australian credit" (Dua & Graff, 2017) and the popular MNIST (Lecun et al., 1998) dataset.

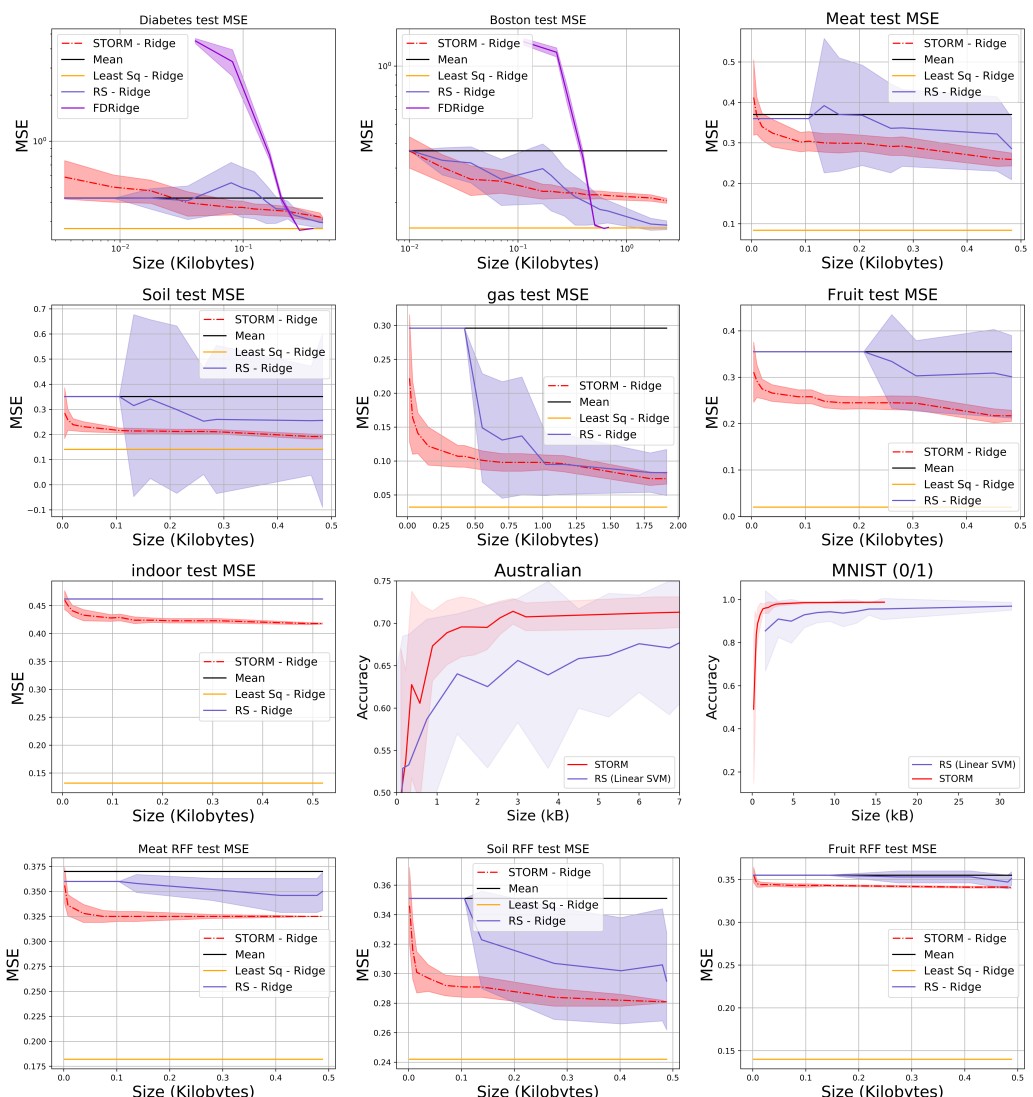

Figure 5: First 7 plots and last row represents MSE vs memory budget for ridge regression. The shaded region represents one standard deviation. The baselines are vanilla least-square ridge regression (Least Sq. Ridge) using SGD, mean of the training set labels (Mean), Random sampling ridge regression (RS - Ridge), and Frequent direction streaming sketch (FD Ridge) (Shi & Phillips, 2021). STORM takes memory less than even a single sample and hence can perform risk minimization where even random sampling cannot possibly help. Some of the high dimension data like Fruit shows this predominantly. The last 2 plots of 3rd row represents classification comparing STORM with random sampling. In last row we feature transform the data using the Random Fourier Transform (Rahimi & Recht, 2008). For each experiment, we scale the number of features 5 times its original size. All results are calculated over 100 runs.

## 6 DISCUSSION

STORM is a scalable framework to solve regularized ERM problems. It is a memory-efficient framework that supports regularization and streaming, as long as the loss function can be described using LSH collision probabilities. The sketch provides compression ranging from $1/50$ to $1/10000$ of the original size and can take size effectively even less than a sample. Additionally, our approach is appealing because it performs well in high dimensions, which can be a prerequisite for feature transformations such as random Fourier features (Rahimi & Recht, 2008). We expect that STORM can be extended to work for important tasks such as kernel classifiers and kernel ridge regression. This will greatly expand the modeling options for applications like distributed learning via ERM on edge devices.

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

# A APPENDIX

## A.1 PROOF OF THEOREMS

**Theorem 1. Unbiased RACE Estimator** *Let $X = \mathcal{S}[i, l_i(\theta)]$. Then*

$$\mathbb{E}[X] = \sum_{\mathbf{x} \in \mathbf{D}} k(\mathbf{x}, \theta) \qquad \mathrm{var}(X) \leq \left( \sum_{\mathbf{x} \in \mathcal{D}} \sqrt{k(\mathbf{x}, \theta)} \right)^2$$

*Proof:* The proof is detailed in Coleman(2019) Coleman et al. (2019) and Luo(2018) Luo & Shrivastava (2018).

**Theorem 3.** *The set of STORM-approximable functions $S_L$ contains all LSH collision probabilities and is closed under addition, subtraction, and multiplication.*

It is straightforward to see that STORM can approximate $\sum_{x \in \mathcal{D}} k(x, y)$ as long as there is an LSH function with the collision probability $k(x, y)$. To prove the theorem, it is sufficient to show that given two LSH collision probabilities $k_1(x, y)$ and $k_2(x, y)$, STORM sketches can approximate the following two functions

$$f_1(y) = \sum_{x \in \mathcal{D}} k_1(x, y) \pm k_2(x, y) \qquad f_2(y) = \sum_{x \in \mathcal{D}} k_1(x, y) k_2(x, y)$$

Note that one can always write a product of (weighted) sums $\left( \sum_n w_n k_n(x, y) \right) \left( \sum_m w_m k_m(x, y) \right)$ as the sum of (weighted) products $\sum_{n,m} w_n w_m k_n(x, y) k_m(x, y)$. Therefore, the previous two situations ensure that the set is closed under addition, subtraction and multiplication.

**Addition and Subtraction:** Because of the distributive property of addition,

$$\sum_{x \in \mathcal{D}} k_1(x, y) \pm k_2(x, y) = \sum_{x \in \mathcal{D}} k_1(x, y) \pm \sum_{x \in \mathcal{D}} k_2(x, y)$$

One can then construct a STORM sketch $\mathcal{S}_1$ for the $k_1(x, y)$ summation and a second STORM sketch $\mathcal{S}_2$ for $k_2(x, y)$ summation. We can estimate any linear combination of $k_1(x, y)$ and $k_2(x, y)$ by with a weighted sum of $\mathcal{S}_1$ and $\mathcal{S}_2$.

**Multiplication:** To approximate sums over the product $k_1(x, y) k_2(x, y)$, we rely on LSH hash function compositions. Suppose we have an LSH function $l_1(x)$ with collision probability $k_1(x, y)$ and $l_2(x)$ with $k_2(x, y)$. Consider the hash function $l(x) = \pi(l_1(x), l_2(x))$ where $\pi(\mathbf{a}, \mathbf{b})$ is an injective (or unique) mapping from $\mathbb{Z}^2 \to \mathbb{Z}$. An example of such a mapping is the function $\pi(\mathbf{a}, \mathbf{b}) = p_1^a p_2^b$ where $p_1$ and $p_2$ are coprime. Since the mapping is injective, this means that $l(x) = l(y)$ only when $l_1(x) = l_1(y)$ and $l_2(x) = l_2(y)$. Therefore, $\Pr[l(x) = l(y)] = \Pr[l_1(x) = l_1(y) \cap l_2(x) = l_2(y)]$
Make the choice of the LSH functions $l_1$ and $l_2$ independently, so that the probability factorizes

$$= \Pr[l_1(x) = l_1(y)] \Pr[l_2(x) = l_2(y)] = k_1(x, y) k_2(x, y)$$

Therefore, one can construct a STORM sketch for the product using the LSH function $l(x) = \pi(l_1(x), l_2(x))$.

**Theorem 4.** *When $p \geq 2$, the PRP collision probability $g([\theta, -1], [\mathbf{x}, y])$ is a convex surrogate loss for the linear regression objective such that*

$$\arg \min_\theta g([\theta, -1], [\mathbf{x}, y]) = \arg \min_\theta \|y - \mathbf{x}\theta\|$$

*Proof:* For the surrogate ERM problem to have the same solution as the linear regression ERM problem, it is sufficient to show two things: that the surrogate loss is convex and that the global minima of the surrogate loss and the linear regresssion loss appear in the same location. The surrogate loss ($\mathcal{L}(\theta, \mathbf{x}, y)$) is

$$g([\theta, -1], [\mathbf{x}, y]) = \frac{1}{2} \left( 1 - \frac{1}{\pi} \cos^{-1}(\langle [\theta, -1], [\mathbf{x}, y] \rangle) \right)^p + \frac{1}{2} \left( 1 - \frac{1}{\pi} \cos^{-1}(-\langle [\theta, -1], [\mathbf{x}, y] \rangle) \right)^p$$

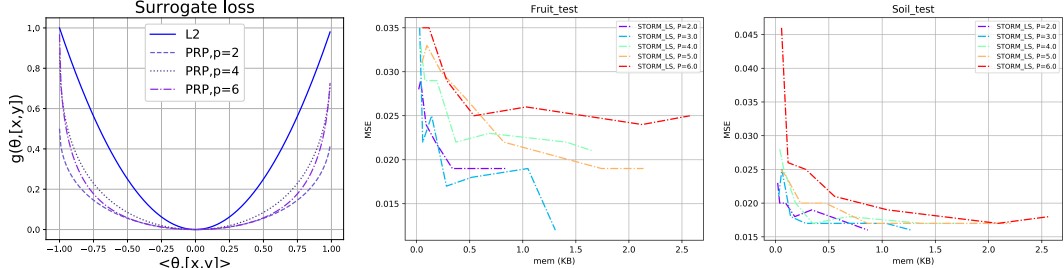

Figure 6: Left: Surrogate regression loss (STORM regression loss) for different values of p. It has the highest convexity near $p = 4$. Right two: Mean square error with varying sketch size with different p on two real world datasets. We observe that p=3,4 works the best.

For the sake of notation, we will put $a = [\theta, -1]$, $b = [\mathbf{x}, y]$, and

$$f(\mathbf{a}, \mathbf{b}) = \left(1 - \frac{1}{\pi}\cos^{-1}(\langle[\theta, -1], [\mathbf{x}, y]\rangle)\right) \qquad \mathcal{L}(\mathbf{a}, \mathbf{b}) = \frac{1}{2}f(\mathbf{a}, \mathbf{b})^p + \frac{1}{2}f(\mathbf{a}, -\mathbf{b})^p$$

We will use the fact that

$$\nabla_a f(\mathbf{a}, \mathbf{b}) = \nabla_a f(\mathbf{a}, -\mathbf{b}) = \frac{\mathbf{b}}{\pi\sqrt{1 - |\langle\mathbf{a}, \mathbf{b}\rangle|^2}}$$

**Location of Minima:** The minimum of the surrogate loss is same as the minimum for least squares linear regression. Using the chain rule, the gradient of the surrogate loss is

$$\nabla_a \mathcal{L}(\mathbf{a}, \mathbf{b}) = \frac{p\left(f(\mathbf{a}, \mathbf{b})^{p-1} - f(\mathbf{a}, -\mathbf{b})^{p-1}\right)}{2\pi\sqrt{1 - |\langle\mathbf{a}, \mathbf{b}\rangle|^2}}b$$

When $p = 1$, the gradient is always zero. When $p \geq 2$, the derivative is zero when $\langle\mathbf{a}, \mathbf{b}\rangle = \langle[\theta, -1], [\mathbf{x}, y]\rangle = 0$ because that is where $f(\mathbf{a}, \mathbf{b}) = f(\mathbf{a}, -\mathbf{b})$. Thus, the surrogate loss has the same minimizer as the least squares loss.

**Convexity:** At index $(i, j)$, the Hessian of the surrogate loss is

$$[\nabla_a^2 \mathcal{L}(\mathbf{a}, \mathbf{b})]_{(i,j)} = \frac{\partial}{\partial a_j}[\nabla_a \mathcal{L}(\mathbf{a}, \mathbf{b})]_i = \frac{\partial}{\partial a_j}b_i\frac{p\left(f(\mathbf{a}, \mathbf{b})^{p-1} - f(\mathbf{a}, -\mathbf{b})^{p-1}\right)}{2\pi\sqrt{1 - |\langle\mathbf{a}, \mathbf{b}\rangle|^2}}$$

Thus, the Hessian

$$\nabla_a^2 \mathcal{L}(\mathbf{a}, \mathbf{b}) = b^\top\nabla_a\left(\frac{pf(\mathbf{a}, \mathbf{b})^{p-1} - pf(\mathbf{a}, -\mathbf{b})^{p-1}}{2\pi\sqrt{1 - |\langle\mathbf{a}, \mathbf{b}\rangle|^2}}\right)$$

The gradient

$$\nabla_a\left(\frac{pf(\mathbf{a}, \mathbf{b})^{p-1} - pf(\mathbf{a}, -\mathbf{b})^{p-1}}{2\pi\sqrt{1 - |\langle\mathbf{a}, \mathbf{b}\rangle|^2}}\right)$$

$$= \left(\frac{p(p-1)f(\mathbf{a}, \mathbf{b})^{p-2}}{2\pi\sqrt{1 - |\langle\mathbf{a}, \mathbf{b}\rangle|^2}}\nabla_a f(\mathbf{a}, \mathbf{b}) + \frac{p(p-1)f(\mathbf{a}, -\mathbf{b})^{p-2}}{2\pi\sqrt{1 - |\langle\mathbf{a}, \mathbf{b}\rangle|^2}}\nabla_a f(\mathbf{a}, -\mathbf{b})\right)$$

$$+ \left(pf(\mathbf{a}, \mathbf{b})^{p-1} - pf(\mathbf{a}, -\mathbf{b})^{p-1}\right)\nabla_a\left(\frac{1}{2\pi\sqrt{1 - |\langle\mathbf{a}, \mathbf{b}\rangle|^2}}\right)$$

Simplifying, we obtain

$$\frac{\mathbf{b^Tb}}{2\pi}\left(\frac{p(p-1)f(\mathbf{a}, \mathbf{b})^{p-2} - p(p-1)f(\mathbf{a}, -\mathbf{b})^{p-2}}{\pi(1 - |\langle\mathbf{a}, \mathbf{b}\rangle|^2)} + \langle\mathbf{a}, \mathbf{b}\rangle\frac{pf(\mathbf{a}, \mathbf{b})^{p-1} + pf(\mathbf{a}, -\mathbf{b})^{p-1}}{(1 - |\langle\mathbf{a}, \mathbf{b}\rangle|^2)^{3/2}}\right)$$

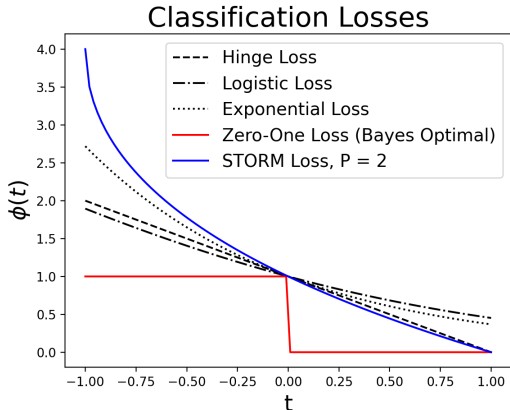

Figure 7: Comparison of various classification-calibrated loss functions, including the STORM classification loss.

Which gives the following expression for the Hessian

$$\nabla_\theta^2 \mathcal{L}(\mathbf{a}, \mathbf{b}) = \mathbf{b}^T \mathbf{b} \rho(\mathbf{a}, \mathbf{b})$$

where

$$\rho(\mathbf{a}, \mathbf{b}) = \frac{1}{2\pi} \left( \frac{p(p-1)f(\mathbf{a}, \mathbf{b})^{p-2} - p(p-1)f(\mathbf{a}, -\mathbf{b})^{p-2}}{\pi(1 - |\langle \mathbf{a}, \mathbf{b} \rangle|^2)} + \langle \mathbf{a}, \mathbf{b} \rangle \frac{pf(\mathbf{a}, \mathbf{b})^{p-1} + pf(\mathbf{a}, -\mathbf{b})^{p-1}}{(1 - |\langle \mathbf{a}, \mathbf{b} \rangle|^2)^{3/2}} \right)$$

for $p \geq 2$ $\rho(\mathbf{a}, \mathbf{b}) > 0$ $\forall \langle \mathbf{a}, \mathbf{b} \rangle \in (-1, 1)$. Hence, the Hessian is positive semidefinite and the function is convex in $\mathbf{a}$. Also note that the function is convex in $\theta$, since the restriction of the last dimension of $\mathbf{a}$ to $-1$ is the restriction of a convex function to a convex set. Refer Figure 6(a).

**Theorem 2.** *Consider labels $y \in \{-1, 1\}$. The loss function $g(\theta, [\mathbf{x}, y])$ for the linear hyperplane classifier is a classification-calibrated margin loss and is STORM-approximable.*

$$g(\theta, [\mathbf{x}, y]) = 2^p \left( 1 - \frac{1}{\pi} \cos^{-1}(-y\langle \theta, \mathbf{x} \rangle) \right)^p$$

*Proof.* First, we show that the loss is classification-calibrated. Then, we show that the loss can be estimated using STORM.

**Loss is Classification-Calibrated:** A necessary and sufficient condition for a convex[2] loss function $\phi(t)$ to be classification-calibrated is for $\frac{d}{dt}\phi(t) < 0$ at $t = 0$. Here, $t = yh(x)$, where $h(x)$ is the model. For a linear hyperplane classifier, $h(x) = \langle \theta, \mathbf{x} \rangle$. The loss is therefore

$$\phi(t) = 2^p \left( 1 - \frac{1}{\pi} \cos^{-1}(-t) \right)^p$$

$\phi(t)$ is convex when $p \geq 2$ for the same reasons discussed in the proof of Theorem 4. Note that the simple asymmetric LSH for the inner product that we have used throughout the paper[3] requires $t \in [-1, 1]$. The derivative is

$$\frac{d}{dt}\phi(t) = 2^p p \left( 1 - \frac{1}{\pi} \cos^{-1}(-t) \right)^{p-1} \frac{-1}{\pi\sqrt{1 - t^2}}$$

---

[2]For non-convex $\phi(t)$, the sufficient conditions are more complicated.

[3]There are other asymmetric inner product LSH functions without this requirement, and in practice one usually scales the data to make the condition true.

At the origin, $\frac{d}{dt}\phi(t)$ is $-1/\pi$. Therefore, the loss is classification calibrated. Figure 7(b) compares our STORM surrogate classification loss against popular margin losses.

**Loss is STORM-Approximable:** Consider the asymmetric LSH function for the inner product where we premultiply $\mathbf{x}$ by $-y$. The collision probability under this LSH function is

$$k(\theta, \mathbf{x}) = \left(1 - \frac{1}{\pi} \cos^{-1}(\langle \theta, -y\mathbf{x}\rangle)\right)$$

as desired. $\qquad\square$

***Theorem 5.*** *Let $\hat{g}(\theta)$ is the median of means estimate on the STORM sketch, with $R$ rows, then with probability more than $1 - \delta$*

$$|\hat{g}(\theta) - g(\theta)| \leq 6\frac{\sum_{\mathbf{x}\in\mathcal{D}} \sqrt{k(\mathbf{x},\theta)}}{\sqrt{R}}\sqrt{\log(1/\delta)}$$

*Proof.* Consider the following axiom-

*Let $Z_1, ... Z_R$ be $R$ i.i.d. random variables with mean $\mathbb{E}[Z] = \mu$ and variance $\leq \sigma^2$. Divide the $R$ variables into $g$ groups so that each group contains $m = R/g$ elements, and take the empirical average within each group. The median-of-means estimate $\hat{\mu}$ is the median of the $g$ group means. If $g = 8\log(1/\delta)$ and $m = R/g$, then the following statement holds with probability $1 - \delta$.*

$$|\hat{\mu} - \mu| \leq 6\frac{\sigma}{\sqrt{R}}\sqrt{\log 1/\delta}$$

This proof is given in Alon et al. (1999) as the proof of Theorem 2.1 (which is a slightly more general version of the statement above). $\qquad\square$

Now, et $Z(q)$ be the median-of-means estimate constructed using the $R$ unbiased estimators of the sketch with $R$ rows. Then with probability $1 - \delta$,

$$|\hat{g}(\theta) - g(\theta)| \leq 6\frac{\sum_{\mathbf{x}\in\mathcal{D}} \sqrt{k(\mathbf{x},\theta)}}{\sqrt{R}}\sqrt{\log(1/\delta)}$$

From Theorem 1, we know that $\sigma \leq \sum_{\mathbf{x}\in\mathcal{D}} \sqrt{k(\mathbf{x},\theta)}$
Substituting this variance bound into the given Lemma proves the theorem.
**Required number of repetitions:** The square root of collision probability $\sqrt{k(\mathbf{x},\theta)} \leq 1$ always. Hence, in the worst case the value of $\sum_{\mathbf{x}\in\mathcal{D}} \sqrt{k(\mathbf{x},\theta)}$ is $N$. For $|\hat{g}(\theta) - g(\theta)| \leq C\sqrt{\log(1/\delta)}$, we need number of sketch rows $R = O(N^2)$.

**SGD convergence and sketch size for hyperplane based method:** Assuming that 1) $\hat{g}(\theta)$ is bounded below, 2) The gradient is L-Lipschiz continuous and 3) the gradients bounded as $\mathbb{E}\left[\|\nabla\hat{g}_i(\theta)\|^2\right] \leq \sigma^2$, with constant step size $\eta$ we will get the error $(\hat{g}(\theta_t) - \hat{g}(\theta^*))$ bounded by $O(1/t) + O(\eta)$.
For a simplistic assumption that we are drawing a new sample every time the total number of steps is equivalent to the number of repetitions $R$ in the sketch. Hence the error at the end of the iteration will be bounded in terms of sketch size by $O(1/R) + O(\eta)$.

## A.2 EXPERIMENTS

### A.2.1 EXPERIMENT ON 2D SYNTHETIC DATASET

We plot the estimated parameter $\hat{\theta}$ from the STORM on 2D synthetic data (Train: 1600 points, Test: 400 points) generated from Gaussian distribution. Refer to Figure 8.

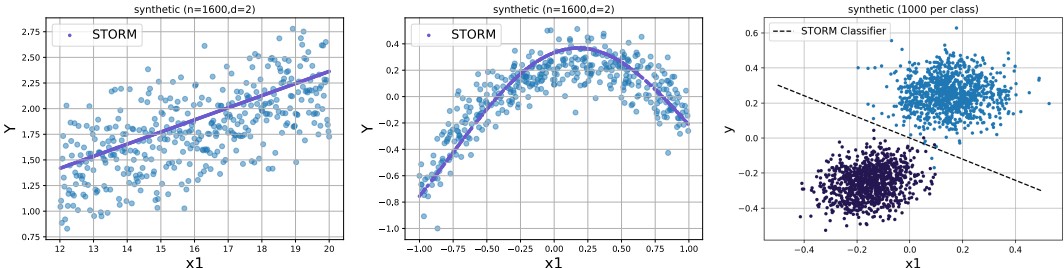

Figure 8: (a) Regression with linear model. (b) regression with linear model and Random Fourier feature Rahimi & Recht (2008) mapping, and (c) classification STORM losses on synthetic 2D datasets.

### A.2.2  NON LINEAR REGRESSION ON SYNTHETIC DATA:

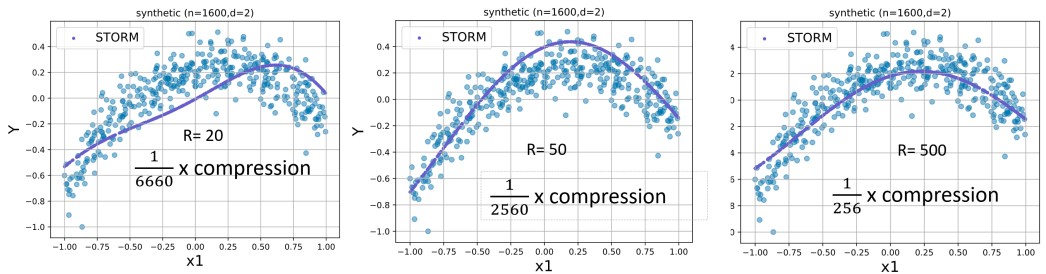

Figure 9: The random Fourier feature maps the input dimension to a higher dimension for linear model training. Here we map it to a 10 dimension space and project back the solution in 2D. The figure shows the learnt model with different STORM sketch sizes. On the given synthetic data, STORM achieves a compression of $1/256$ times to $1/6660$ times.

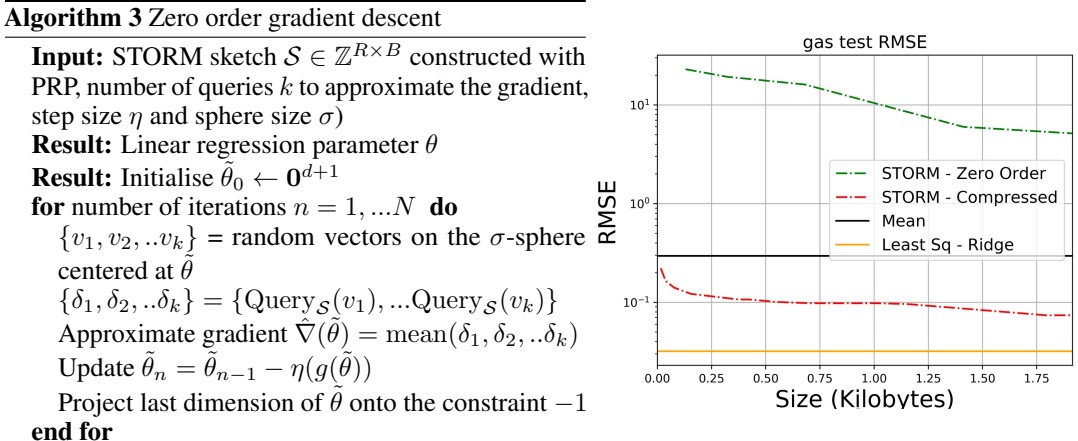

Figure 10: Left: Shows the algorithm for zero order gradient descent. Right: The compressed PRP sketch (which uses Hyperplane based gradient descent) is $Rp$ bits in size. And on a experiment with real data we found out that it can give acceptable model with a very less sketch size.

### A.2.3  COMPARISON WITH ZERO ORDER GRADIENT DECENT

We perform derivative-free optimization Conn et al. (2009) here on the original STORM sketch. Here, the black-box access to the loss function of interest (or its sharp approximation) is sufficient. For the descent algorithm, we initialize $\tilde{\theta}$ to zeros in $d+1$ dimensions. The additional dimension

is due to querying the sketch with $[\theta, -1]$ rather than $[\theta]$. We compute the approximate gradient by querying the sketch with equidistant points in a ball around $\tilde{\theta}$ and update the parameter. After each iteration, we project the last dimension of $\tilde{\theta}$ back onto the constraint $\tilde{\theta}_{d+1} = -1$. This takes $R \times B$ size, with an additional factor of sketch cell bit precision. Hence it is more memory intensive than the hyper-plane based stochastic gradient descent algorithm. Refer the figure below.

### A.2.4 NON-REGULARISED LINEAR REGRESSION

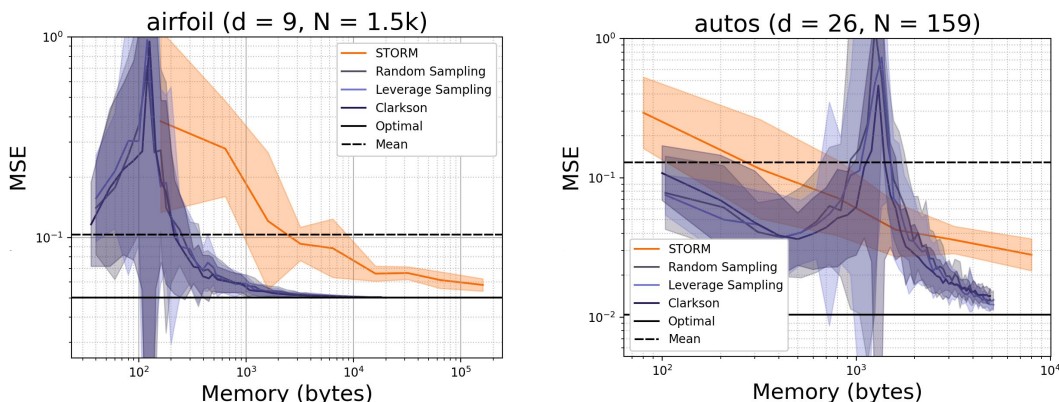

Figure 11: Here, we report the mean square error for our method without any regularization, when compared with baselines at a variety of memory budgets.

In Figure 11, we observe a double-descent phenomenon for our sampling baselines, explaining the peak near the intrinsic dimensionality of the problem. This sample-wise double descent behavior was recently proved for linear regression by Nakkiran (2019). STORM does not experience the double descent curve in practice because the entire dataset (not just a subsample) is used to minimize the loss. We perform favorably against baselines in memory regimes affected by double descent and STORM performs competitively in other memory regimes. We also observe that the $\theta$ found using STORM converges to the optimal $\theta$ under least-squares ERM. The datsets in use are taken from UCI repository.

