# OpenReview forum: "STORM: Sketch Toward Online Risk Minimization"
_ICLR.cc/2022/Conference — ICLR 2022 Submitted_

### Official Review · Reviewer_8LGD · 2021-11-01

**Correctness:** 3
**Technical Novelty And Significance:** 3
**Empirical Novelty And Significance:** 3
**Recommendation:** 6
**Confidence:** 4

**Main Review:**

Strengths:
-Impact: The proposed flexible method reduces the scale of data set and the dependence of model training on hardware performance, which may be helpful for distributed learning in the future.
-Experiment: The author has done relatively sufficient comparative experiments, including classification and regression under linear and nonlinear conditions with several datasets.
-Writing: Overall, the manuscript is well-written and addressing a relevant problem by proposing an interesting learning method suited to edge devices.
-Reproducibility: Key resources (e.g., proofs, code, data) are available and key details (e.g., proofs, experimental setup, parameters selection) are sufficiently well-described for competent researchers to confidently reproduce the main results.

Weekness:
-Experiment: Important details missing, which can help the paper quality significantly. Including the MSE and accuracy criteria, time complexity and space complexity of different methods should also be discussed. During inference, is the proposed approach less computational expensive than baseline approaches? It would be worth reporting the run-time (e.g in seconds) or computational complexity or number of learnable parameters.
-Parameters: The proposed approach STORM is heavily based on some of the hyper-parameters, for instance, p, there is no empirical evidence that how the performance of the model will vary by changing p. Besides, the value of p is different for different datasets. Ablation experiments are helpful to validate the robustness of the selected value for p instead of the two regression experiments for discussing P values with similar datasets in Fig. 6
-Description: In general, there is a lack of detailed explanation of replicates and statistical methods used in the study.
-Performance: Although this edge computing method reduces the dependence on hardware, the reduction of accuracy seems to have a great impact from the experimental results in this paper. Can it be compared with more similar methods to highlight the advantages of this paper? Some results in Figure 5 are worse than the benchmark method. Is it necessary to have some discussion?

**Summary Of The Paper:**

In this paper, the authors focus on reducing the cost of conventional ERM training and propose a new framework, STORM, which integrates the approaches of locality sensitive hashing and sketches. Besides, the authors use specific optimization algorithms to solve that this framework can’t be directly applied to popular classification and regression tasks. The experimental results prove that the proposed procedure can achieve better results compared to the baselines in the operational environment of low performance and low memory.

**Summary Of The Review:**

The paper contributes some new ideas to reduce the compuation complexity of ERM. The experimental evaluation on STORM is adequate, and the results convincingly support the computaion strategy  of locality sensitive hashing and sketches.

---

### Official Review · Reviewer_7LQM · 2021-11-01

**Correctness:** 4
**Technical Novelty And Significance:** 1
**Empirical Novelty And Significance:** 1
**Recommendation:** 3
**Confidence:** 3

**Details Of Ethics Concerns:**

No concern

**Main Review:**

# Quality and Clarity

The paper is clearly written, with very few typos (listed below). Nice intuitions are regularly given, and I especially liked the "Intuition" paragraph in Section 3. Good writing!

The problem of approximate ERM with low memory is theoretically compelling, and I _like the idea_ of finding approximate ERM solutions using less memory than a single sample, but I don't understand when this memory constraint is actually realistic? When can a machine compute hash functions for the all data, but cannot store a single sample (or $O(1)$ samples) in memory? I ask because the experiments are not evidently compelling, but they do use very very little space. So, when is this extreme setting compelling?

# Theoretical Basis

There are some odd points in the theory given behind their algorithm and data structure:
- Other than using non-standard hashing functions, what distinguishes the STORM sketch from the RACE sketch? They seem the exact same to me.
- Theorem 4 shows that the STORM optimum recovers the Least Squares optimum for the linear regression problem. You also use STORM for classification; do you have any theory to justify why STORM should work for classification?
- Theorem 5 has this $\sum_{x \in \mathcal D} \sqrt{k(x,\theta)}$ term. How should I interpret this? It is just the standard deviation of the RACE sketch? Why not just upper bound this by $N$ in the numerator? When is this numerator much smaller than $N$?
  - Also, is this reuslt
- Theorem 5 mentions the Median-of-means estimate, which is formally defined in the appendix, but I would recommend adding it into the body of the paper. Why is estimator mentioned, if it doesn't seem like you use it in the experiments or optimization? Do you use median-of-means in the optimization or experiments?
- Is the choice of $\tanh$ on page 6 totally heuristic?

My *review of the theoretical results* and data structure design is that the results are believable and seem correct, but lack technical novelty. Some of this (e.g. Thm 5) is not obviously distinct from prior work (e.g Thm 1's variance). Some of what I would like to see is not shown (e.g. correctness of using $\tanh$, classification version of Thm 4). Weak overall.

# Experimental Evidence

The experimental evidence continues to be a bit confusing. I don't really understand why this is showing that STORM is a good model in practice. There are 12 plots showing statistical error vs. amount of memory used, but very few of these plots tell a compelling story:
_In the following, I refer to plots in the left-to-right then top-to-bottom order, so the "4th plot" refers to the first plot on the second row._
- Plots 2, 3, 4, 5, 6, 7, 9, 10, 11, and 12 suggest that as memory capacity increases, STORM does *not* approach the true solution. Instead, the variance gets small while the bias remains large.
- Plots 3, 4, 6, 7, 10, and 12 even further show that STORM returns a model whose error is closer to a naive mean than an actual Least Squares solution.
- Plots 8 and 9 don't have a benchmark to show what is achievable with unbounded memory. Plot 9 is fine, since near-perfect accuracy is possible, but plot 8 sees maximum accuracy 0.75, and it's not obvious this is competitive with a larger memory model.
- Plots 1 and 2 don't have legible y-axes. There's too few written ticks, and for some reason only these two plots are on a log/log scale.

So, it seems that experimentally, STORM uses very little space to be slightly better than computing a naive mean of the data. I don't see how this is compelling, especially since I'm not clear why the restriction on the memory has to be so intense. As far as I am aware, 1 Mb of data is very very cheap today, so what's the utility of working on the order of kilobytes if the error of the model is so huge?

_My conclusions / overall review is in the "Summary of the Review Section"_

---
## Minor Notes
- Use `\citep{...}` instead of `\cite{...}` when adding a citation that doesn't naturally flow in the grammar. This is (e.g.) most of the citations you use in the introduction.
- [Page 1, end of "Linear Sketches" paragraph] Why is it that sketching cannot accommodate regularization? If we can approximate $\|Ax-b\|$ for all $x$, then we can also approximate $\|Ax-b\| + \|x\|$, right?

1. [Page 1, end of first paragraph]: The "Bill Allcock (2001)" citation seems weird. The citation in the references also seems formatted wrong. Should this be "Breshanan et al"?
1. [Page 3, first line of section 2.2]: It should be "A locality sensitive hash (LHS) function" not "A Locality sensitive hashing function" (capitalization, and "hash" is more common than "hashing" but that one's personal preference)
1. The Section 3 Intuition block is written _really_ nicely. Props.
1. [Page 4, bottom equation]: $p$ isn't defined, and it's not intuitive at all that some parameter should be floating in the exponent.
1. [Page 5, Thm 2 equation]: Same as above
1. [Page 6, in the sentence "We now have the..."]: The left $\|$ symbol is missing in the inline math.
1. [Page 7, Thm 5]: Should be "Let $\hat g(\theta)$ be the" instead of "Let $\hat g(\theta)$ is the"
1. [Page 8, end of Figure 5 caption]: Specify _here_ that you increase the feature space to attempt to linearize the problem, not just to introduce spurious dimensionality (which is kinda how it reads as is)
1. [Page 9, Table 1 caption, last sentence]: "The dimensions in brackets" not "The dimension in bracket"

**Summary Of The Paper:**

The paper considers approximately computing ERM problems for real data, with an emphasis on regression and classification, in the *very low space* regime (most experiments are below 1Kb). This is done my maintaining a standard locality sensitive hashing table, but with specialized hash functions that allow for computing a surrogate for common regression and classification ERM objectives.

Due to the extreme space constraint, optimization is done with a heuristically characterized SGD. A significant amount of experiments is presented.

**Summary Of The Review:**

The paper has weak theoretical and experimental components. The theory doesn't obviously show anything new of sufficient generality. The data structure appears to be the same as in prior work on LSH. The experiments suggest that using the STORM estimator is only slightly better than returning the mean of your data.

I don't see the strengths of this paper anywhere. I got more and more confused about this paper the more I read.

It's well written, but I don't see the substance beneath the nice writing.

---

> ### Author Response · Authors · 2021-11-23
> **Thank you for your comments and suggestions.**
>
> Thank you for the helpful and detailed reviews. Following are our comments.
>
> Comparison with RACE: : The work by (Coleman & Shrivastava, 2020) -RACE and STORM are not same. They both use a sketching framework that is inspired from count-min-sketch and Locality sensitive hashing. Unline any existing method our work  proposed a new convex loss (PRP and CRP) for regression and classification which can be well approximated with the STORM sketch (motivated by Coleman & Shrivastava, 2020). Additionally we talk about the challenge of performing optimization over the sketch query, which is a new (and non-trivial) problem not previously considered in the literature. Because standard zero-order (black box) optimization methods do not solve the problem well (check Appendix A.2.3- added recently), we introduce our hyperplane based optimization method (due to limited space we added the comparison in appendix).
>
> Theorem 2 has defined the CRP and the classification calibration is proved in the appendix Theorem 2 proof. Just like the Theorem 4 proof, Theorem 2 proof is given to ensure a working classification loss using STORM sketch. Also refer to Figure 4(c) or Figure 7(appendix).
>
> Please refer to Theorem 5 in the revised version.
>
> Tanh: We use tanh as the smooth approximation of the sign() function. There is no single best approximation. Sigmoid function can also be used in this case. However tanh provides faster convergence by having stronger gradients ( Efficient Backprop by LeCun et al.), hence it was selected in our experiments. The Hyperplane based gradient descent provides a way to convert the given ERM task to classification task on a super compressed and private sketch. This classification task on sketch can be done via any off the shelf classifier.
> STORM uses very little space and even less than a sample sometimes is a huge compression award. 1Mb of data is very cheap today, however an edge device today runs many (100s and even thousands of) applications in parallel, making it harder to achieve several models training in limited space.

---

> > ### Comment · Reviewer_7LQM · 2021-11-23
> > **What about the experiments?**
> >
> > Thanks for your reply. I'll take some time to look into these points to see if they convince me to strengthen my review of the theoretical contributions. However, I note that your response **fails to acknowledge any concerns about your experiments**.
> >
> > Could you comment on the experimental evidence? My impression is that the experiments are very weak, and do not suggest that STORM is an effective method, even if it uses low memory.
> >
> > I even have "Experimental Evidence" as a section with a large bold header....
> >
> > Thanks.

---

> > > ### Comment · Reviewer_7LQM · 2021-11-29
> > > **No response from the authors**
> > >
> > > The discussion period is ending today and the authors have not responded to my many questions about the experiments, so my score remains unchanged. It's been 6 days since I followed up, so they had time to respond.
> > >
> > > This paper should clearly be rejected. My score is unchanged.

---

### Official Review · Reviewer_yRcK · 2021-11-01

**Correctness:** 3
**Technical Novelty And Significance:** 3
**Empirical Novelty And Significance:** Not applicable
**Recommendation:** 5
**Confidence:** 4

**Main Review:**

The paper proposes an online sketching algorithm (STORM) for empirical risk minimization(particularly for linear regression loss functions), which only needs to save the integer count values in the sketch. Particularly, it uses the local-sensitive hash functions to divide the space into several regions and save the integer counts values over the regions for the data points. Then, the author shows that how the count values related to a class of the surrogate losses for the $\ell_2$ loss and how to evaluate a hypothesis $h_\theta$ using such sketch, and how to optimize the $\theta$. Finally, the author gives an empirical evaluation with ridge regression and classification.

Overall, I think the method the paper proposes is interesting and the author provides the corresponding analysis. The paper is well-written and easy to follow up. The empirical results also show the advantage of this method.  However, I have the following concerns/questions:

(1) The conclusion section and the experiments are focusing on the regularized ERM problems, while most of the theoretical analysis seems to be for the non-regularized version. Is the analysis still hold for the regularized ERM case?

(2) The related-work section of the linear sketch seems to be incomplete and inaccurate. In my understanding, the Sketch-and-Solve technique also holds for the ridge regression for the case when $n >> d$, see, e.g., the Theorem~16 of [1](this paper also studies the case when $n << d$, but it seems can not be supportive for the streaming setting). I think it should be added to the experiments as a baseline, if I do not have some misunderstandings.

(3) I am a little confused about the theoretical advantage of this method. In my understanding, for the case when $n >> d$ the linear sketch can get a good approximation in roughly $O(d^2)$ space, while STORM needs $O(N^2)$. Hence the advantage of the STORM is for the case when $n << d$. Is the above statement correct?

(4) I am curious about the empirical comparison for the non-regularized version, and I think it can be a good supportive part(not necessary), because the theoretical analysis also holds in this case, and we will have more baselines at this time.

Reference:

[1] Haim Avron, Kenneth L. Clarkson, David P. Woodruff. Sharper Bounds for Regularized Data Fitting.

**Summary Of The Paper:**

The paper proposes an online sketching algorithm (STORM) for empirical risk minimization(particularly for linear regression loss functions), which only needs to save the integer count values in the sketch. The author also gives an empirical evaluation with ridge regression and classification.

**Summary Of The Review:**

As stated in the previous section, I think the method this paper proposes is interesting, while there are still some perspectives that can be further improved. Hence currently I tend to vote a 5 for this paper. I am willing to raise the score if I have some misunderstandings or some of the concerns can be addressed.

---

> ### Author Response · Authors · 2021-11-23
> **Thank you for your comments and suggestion.**
>
> Thank you for the helpful and detailed reviews. Following are our comments.
>
> The analysis holds for the regularized as well as non-regularized case. This is because our method approximates the loss and optimizes the model to minimize the loss approximation. Arbitrary regularization terms may be added during the optimization process. This stands in contrast to several existing sketch-based methods, which compute the solution to the ERM problem in a way that does not permit the addition of a regularization term after sketching.
>
> It is true that our asymptotic bounds are worse in the N >> d case. However, we would like to point out that the O(N^2) term comes from the very loose upper bound on the sum \sum_{x\in D}\sqrt{k(x,\theta)} < N. Since k(x,\theta) is the loss (scaled to between 0 and 1), the worst-case O(N^2) situation only happens for models where every point in the dataset has maximum loss. Practical modeling situations typically have a substantial fraction of points for which the training loss is small, so we do not see the O(N^2) issue.
>
> We have included a comparison on the non-regularised version. Due to space constraint we have referred to this in Appendix. Please refer to Appendix A.2.4.

---

> > ### Comment · Reviewer_yRcK · 2021-11-28
> > **Thanks for the Feedback**
> >
> > Thank you for the detailed answers and updated papers. Can you continue to clarify the following questions?
> >
> > 1. It says that "This stands in contrast to several existing sketch-based methods, which compute the solution to the ERM problem in a way that does not permit the addition of a regularization term after sketching". But in my understanding, the sketch-and-solve technique can also handle the least square problem with $\ell_2$-regularization. Is it correct?  And I think the sketch-based method can also be seen as "approximates the loss and optimizes the model to minimize the loss approximation".  As the reviewer 7LQM suggests that If we can approximate  $|Ax - b|$ for all x, then we can also approximate $|Ax - b| + |x|$?
> >
> > 2. Thanks for the additional experiments, but this section seems not to give a description of the baselines. Does the method "Clarkson" mean the sketch-and-solve method and the method "leverage sampling" means sample the points according to their leverage scores(which I think can not be adapted to the streaming model but can still be a baseline)? And the experiment results suggest that the STORM will be better than other methods in the range when a double-descent phenomenon happened while it will be worse in other ranges when the space is small or large. Is it right?

---

### Official Review · Reviewer_MTPW · 2021-11-03

**Correctness:** 2
**Technical Novelty And Significance:** 2
**Empirical Novelty And Significance:** 2
**Recommendation:** 5
**Confidence:** 3

**Main Review:**

Here is the detailed comments.

----------------------------pros
1. Overall, I think this paper is well-written.
2. In experiments, the proposed method STORM achieves promising performance although the datasets and models are simple.
3. Theoretical analyses are provides to justify the use of sketching to estimate the surrogate loss with high probability.

----------------------------cons
1. The technique novelty is limited since previous work (Coleman & Shrivastava, 2020) has used LSH to approximate kernel density estimation on streaming setting.
2. For the proposed STORM, there are so many approximations --- sketching be LSH, surrogate loss function approximate the original loss, and heuristic approximate optimization problems. Besides, it is specialized for the linear model. Thus, these features makes it complex and not general for other problems and models.
2. Why to prefer an asymmetric LSH over the symmetric one? Please give more specific explanations in the online learning setting.
3. Why to use these surrogate loss function? Although authors give intuitive explanation, more formal theoretical supports might be better. Moreover, in my opinion, the proposed surrogate loss function can also be used in the original data without sketching. Experiments can be done to see whether it is specified for this problem and can justify the effectiveness and efficiency of LSH sketching more fairly by comparing them.
4. This paper lacks theoretical analysis about generalization and the convergence property of the heuristic optimization algorithms.
5. It also lacks the experimental results in the streaming setting w.r.t. the time step.

----------------------------Minor comments

This paper seems a bit rushed since there are many typos. For example, in Sec. 2.1, the head of the second line should not be $\mathcal{D}$.  Besides, the reference format seems strange.

**Summary Of The Paper:**

This paper considers the online risk minimization setting. To reduce the memory cost and accelerate training, this paper proposes a new sketching algorithm (i.e. STORM) based on the locality sensitive hashing (LSH) technique, which mainly includes two steps -- sketching and training. In particular, it provides two STORM surrogate losses for linear regression and classification, respectively. Besides, it also provide theoretical analysis to justify the use of sketching to estimate the surrogate loss with high probability. Experimental results illustrate the effectiveness and efficiency of STORM over other baselines.

**Summary Of The Review:**

Overall, I think, while the novelty is limited, the proposed method is complex with many non-clear approximations, which makes it not general.

---

> ### Author Response · Authors · 2021-11-23
> **Thank you for your comments and suggestion.**
>
> Thank you for the helpful and detailed reviews. Following are our comments.
>
> Comparison with RACE: We would like to point out that the STORM sketch is motivated from the work RACE by (Coleman & Shrivastava, 2020), which we have cited and explained in section 2. Unlike RACE our work is not proposing approximate kernel density estimation, rather it proposed a new convex loss (PRP and CRP) for regression and classification which can be well approximated with the STORM sketch (motivated by Coleman & Shrivastava, 2020). We provided a solution against the challenge of performing optimization over the sketch query, which is a new (and non-trivial) problem not previously considered in the literature. Because standard zero-order (black box) optimization methods do not solve the problem well (check Appendix A.2.3- added recently), we introduce our hyperplane based optimization method.
>
> We focus on linear models because they allow us to explicitly analyze the loss in terms of well-understood tasks (linear regression and classification). However, the main technical insight of our paper is that a small collection of histograms over the dataset are sufficient to learn a model when the histogram partitions are chosen properly. For example, if the histogram partitions are chosen via the PRP hash function then we can do linear regression. For linear classification, we use the CRP hash to draw the partitions. For non-linear tasks, the partitions may be transformed either by projection with random Fourier features or by using non-linear hash functions. However, the non-linear hash functions are difficult to analyze mathematically.
>
> ALSH: Asymmetry is required to construct hash functions whose collision probabilities are not positive semi-definite kernels. It should be noted that existing work (Coleman & Shrivastava, WWW 2020) and (Runze et. al. KDD 2021) is limited to PSD kernels due to their use of symmetric hashes.
>
> The PRP loss functions is proved to be convex, with the pointwise minima occurring at the optimal model. Similarly CRP is proved to be classification calibrated. Refer to Appendix for the proof of Theorem 2 and 4.

---

### Decision · Program_Chairs · 2022-01-20

**Decision:**

Reject

**Comment:**

The paper proposed a sketching algorithm for empirical risk minimization (ERM) for linear regression and classification. The technique is based on LSH with non-standard hash functions. The reviews indicate that the paper is well written and easy to follow. However, there are several concerns raised regarding its quality. A major one regards the novelty of the paper.

MTPW: “The technique novelty is limited since previous work (Coleman & Shrivastava, 2020) has used LSH to approximate kernel density estimation on streaming setting.“, 7LQM: “My review of the theoretical results and data structure design is that the results are believable and seem correct, but lack technical novelty.” “Other than using non-standard hashing functions, what distinguishes the STORM sketch from the RACE sketch?”

An additional concern is a claim of weak experimental evidence. There seems to be a need for more thorough experiments isolating different components rather than the system as a whole, and in addition the bottom line results provide only a slight lift over a naive baseline (7LQM:  “The experiments suggest that using the STORM estimator is only slightly better than returning the mean of your data.”).

Whether it is the case that the techniques should be improved or that these concerns could be addressed by improving the presentation of the paper, the conclusion is that the paper now is not ready to be published.